# Phase-Aware KANGaussian : Phase-Regularized 3D Gaussian Splatting with Kolmogorov-Arnold Network

## Abstract

Vanilla 3D Gaussian Splatting (Kerbl et al. (2023)) struggles with modelling high frequency details, especially in unbounded scenes. Recent works such as Scaffold-GS (Lu et al. (2023)) and Spec-Gaussian (Yang et al. (2024)) have made tremendous improvements to the reconstruction quality of these high frequency details, specifically in synthetic and bounded scenes, but still struggle with unbounded real world scenes. Therefore, we propose Phase-Aware KANGaussian, a model building on these earlier contributions to produce state-of-the-art reconstruction quality for unbounded real world scenes with greatly improved high frequency details. Phase-Aware KANGaussian introduces a novel phase regularization method that optimizes models from low-to-high frequency, dramatically improving the quality of high frequency details. Phase-Aware KANGaussian is also one of the first few papers to integrate a Kolmogorov-Arnold Network (KAN) into the Gaussian Splatting rendering pipeline to verify its performance against the Multilayer Perceptron (MLP). All in all, Phase-Aware KANGaussian has three main contributions: (1) Introduce a Gaussian Splatting model with state-of-the-art performance in modelling real-world unbounded scenes with high frequency details, (2) a novel phase regularization technique to encode spatial representation and lastly, (3) first few to introduce a KAN into the Gaussian Splatting rendering pipeline.

## 1 Introduction

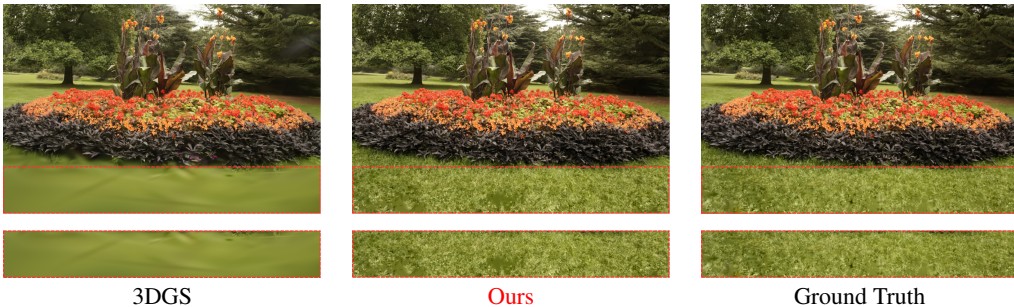

3DGS      Ours      Ground Truth

Figure 1: **Visualization of high frequency detail quality in Phase-Aware KANGaussian** Phase-Aware KANGaussian displaying superior renders of high frequency patches compared to vanilla 3DGS.

3D reconstruction with accurate geometry and lighting has always been the holy grail for 3D computer vision scientists. While traditional methods using photogrammetry were able to achieve considerable success in achieving novel view synthesis, the introduction of Neural Radiance Fields (NeRF) (Mildenhall et al. (2020)) and later 3D Gaussian Splatting (3DGS) (Kerbl et al. (2023)) provide alternatives that promise superior reconstruction quality. The key difference between NeRF and 3DGS is that the former learns an implicit, in contrast to 3DGS's explicit representation of the scene. Comparing their reconstruction quality, 3DGS lags slightly behind NeRF especially in high frequency details, but offers significantly shorter training and inference times, making it the preferred

choice for real time rendering. Some recent works (Lu et al. (2023); Yang et al. (2024); Ye et al. (2024)) have achieved remarkable success in improving the reconstruction quality of high frequency details for 3DGS all while maintaining the real-time rendering capabilities inherited from 3DGS, but the improvements are mostly limited to synthetic and bounded scenes. Therefore, building on the triumphs of earlier works, Phase-Aware KANGaussian aims to extend the quality enhancements to real world unbounded scenes.

In **Phase-Aware KANGaussian**, we employ Kolmogorov Arnold Networks (KANs) in the rendering pipeline in contrast to earlier works (Lu et al. (2023); Yang et al. (2024)) which instead opted for the traditional Multilayer Perceptron (MLP). The rationale behind this decision is to introduce KANs into 3DGS similar to how KANs were successfully integrated into the NeRF (Delin Qu (2024)), and also assess the performance of KANs against MLPs as KANs were claimed to potentially have a faster inference speed (Liu et al. (2024)) than their MLP counterpart. Furthermore, as specular highlights are highly sensitive to viewing angle, KANs have a locality property which can be exploited to achieve superior reconstruction quality, which will be explored in depth in section 3.1.2.

Another key contribution introduced in Phase-Aware KANGaussian is a novel phase regularization technique to achieve low-to-high frequency training. It involves filtering before computing a regularization term in Fourier domain Fuoli et al. (2021); Zhang et al. (2024), and as training progresses, the filter would gradually expand to allow more frequencies to pass, effectively achieving low-to-high frequency regularization. For Phase-Aware KANGaussian, we only consider the phase of the Fourier transform in computing the regularization term as it encodes the spatial relationship and structures in the image, and thus contain more information than the amplitude component.

We validate the effectiveness of our method by focusing on the reconstruction quality of unbounded real world datasets, measured by peak signal-to-noise ratio (PSNR), structural similarity index measure (SSIM) and learned perceptual image patch similarity (LPIPS).

## 2 RELATED WORK

### 2.0.1 ANCHOR-BASED 3D GAUSSIAN SPLATTING.

Anchor-Based 3DGS is a variation of 3DGS first introduced by Scaffold-GS (Lu et al. (2023)), which dramatically improved the geometry and render quality of 3DGS models. The technique involves initializing a sparse voxel grid from the point cloud generated from structure from motion (SfM), and placing an anchor gaussian at the centre of each voxel. The anchor gaussians would then spawn new gaussians — each with their own parameters — around their vicinity. The offsets between the anchor gaussians and these spawned gaussians is learnable, and the parameters of these spawned gaussians are derived by passing a latent feature vector (stored by the anchor gaussian) into a MLP decoder. As the training progresses, the voxel grid is densified with new voxels and anchor gaussians where necessary.

This idea has inspired other works such as Spec-Gaussian (Yang et al. (2024)), which introduced Anisotropic Spherical Gaussians (ASGs) (Xu et al. (2013)) into the model by changing the offset gaussians from spherical gaussians to ASGs. This replacement improved the specular details in the scene, as ASGs are more expressive than spherical harmonics in modelling the view-dependent color function of the gaussians. The work also proposed a coarse-to-fine training method by starting training with down-sampled images, before gradually up-sampling as training progressed.

While these ideas undoubtedly boosted render quality, the improvements are mostly limited to synthetic bounded scenes. For real world scenes, these models still struggle, especially with patches with high frequency details where geometric structure and textures are less predictable. As such, a low-to-high frequency training pipeline would ensure that low frequency details are prioritized by anchor gaussians while high frequency details are left to the offset gaussians, which proves to generate better renders.

### 2.0.2 REGULARIZATION IN 3D GAUSSIAN SPLATTING

Various regularization techniques have been employed to improve the quality of 3D Gaussian Splatting (3DGS) reconstructions, with depth regularization being a notable example (Li et al. (2024); Chung et al. (2024)). This approach typically involves computing a loss between the rendered depth

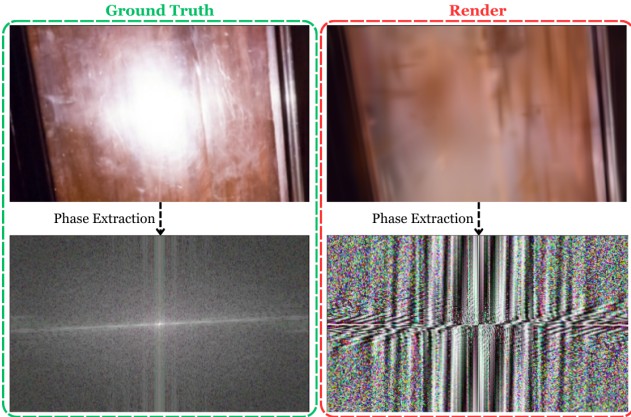

Figure 2: **Phase Discrepency of Specular Detail Renders.** The render fails to capture the phase information of the scene, motivating our phase-regularization technique.

map and the ground truth. In real-world scenes, where ground truth depth maps are often unavailable, monocular depth estimators are commonly used as substitutes. However, specular highlights in these scenes can introduce significant errors in monocular depth estimation, making depth regularization unsuitable for scenes with abundant specular details. To address this limitation, Phase-Aware KANGaussian, designed specifically for enhancing specular renderings, looks to an alternative approach: regularization in frequency domain, which leverages both spectral and spatial loss to improve reconstruction accuracy.

Combining spectral and spatial loss to improve render results has proved successful for various computer vision models (Fuoli et al. (2021); Zhang et al. (2024)). More recently, this technique has also been applied to 3DGS by FreGS (Zhang et al. (2024)) and delivered promising results. In FreGS, mid and high frequency is first removed from the ground truth and render image with a low-pass filter before a regularization term is computed by taking the discrepancy between the amplitude and phase of the resultant images. After a threshold iteration, another band-pass filter would be applied to the frequencies removed by the low-pass filter to compute a similar regularization term for the mid to high frequencies. The band of frequencies allowed through the band-pass filter will be progressively expanded to allow more frequencies to pass, eventually allowing the entire range of frequency to be included in the regularization.

Phase-Aware KANGaussian primarily concentrates on phase information, which encodes the scene's geometry and spatial structure. This focus particularly benefits specular scenes, where high-frequency components like specular highlights can distort frequency-based metrics, which is illustrated in figure 2 where we observe a significant difference in phase of the render and ground truth. By isolating the phase term, the model can optimize the geometry more effectively, preserving the structural integrity of the scene while minimizing the impact of intensity variations caused by specular reflections. This targeted approach leads to more accurate geometric reconstruction, even under challenging lighting conditions.

## 3 PROPOSED METHOD

### 3.1 PRELIMINARIES

#### 3.1.1 3D GAUSSIAN SPLATTING (3DGS).

3DGS is a 3D reconstruction method that represents the scene explicitly using spherical gaussians. Each spherical gaussian is parameterized by a centre $\mathbf{x}$, covariance matrix $\boldsymbol{\Sigma}$ and opacity $\sigma$. Additionally, the view-dependent color of the gaussians are represented by the first three orders of spherical harmonics. To render the spherical gaussians onto the image plane, a tile-based rasterization approach is used. Put simply, the 3D gaussians are converted into 2D gaussians that are parallel to the image plane (Zwicker et al. (2001)) before $\alpha$-blending is applied to each pixel to derive the

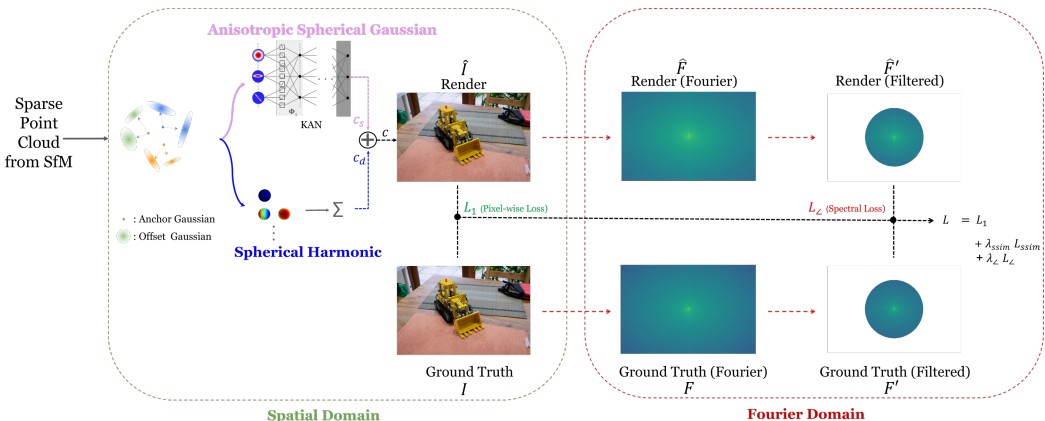

Figure 3: **Overview of Phase-Aware KANGaussian.** Phase-Aware KANGaussian begins with a sparse point cloud initialized from Structure from Motion (SfM). The sparse point cloud will guide the generation of a sparse voxel grid, with an anchor gaussian at the centre of each voxel. These anchor gaussians will then spawn a set of offset gaussians, each with their own set of parameters (scale $s$, rotation $\Sigma$, opacity $\sigma$ and color $c$). The color $c$ of the gaussians are decomposed into a diffuse component $c_d$ and specular component $c_s$, which are computed separately from spherical harmonics and anisotropic spherical gaussians respectively. Once the rendered image $\hat{I}$ is obtained, a pixel-wise loss $L_1$ is obtained from $\hat{I}$ and $I$. Then, Fourier transform is applied to both render and ground truth images to extract frequency information before a filter is applied to narrow the frequency range. A spectral loss is then computed from the filtered render $\hat{F}'$ and filtered ground truth $F'$ using the discrepancy between their phase. As training progresses, the filter will expand linearly with training iterations to allow more frequencies to pass, eventually allowing the entire range of frequencies to be included in the training process.

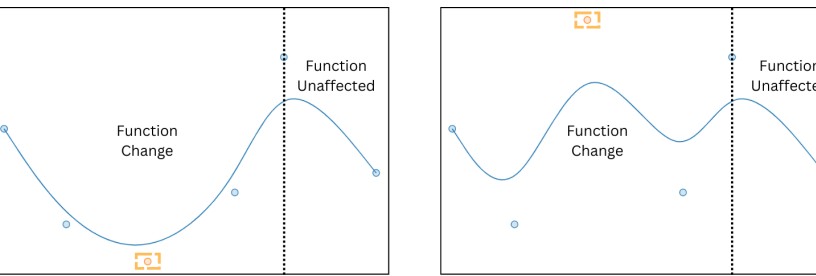

Figure 4: **Visualization of B-Spline Locality Property.** B-Spline locality property, where changes to orange nodes only affect the function in the vicinity

final RGB values:

$$C(\mathbf{P}) = \sum_{i}^{N} c_i \alpha_i \prod_{j}^{i-1} (1 - \alpha_j), \ \alpha_i = \sigma_i \exp\left(-\frac{1}{2}(\mathbf{P} - \mu_i)^T \Sigma'(\mathbf{P} - \mu_i)\right) \tag{1}$$

where $c_i$, $\alpha_i$ and $\mu_i$ are the color, opacity and projected centre of the 2D gaussian respectively.

### 3.1.2 KOLMOGOROV ARNOLD NETWORK (KAN).

KANs are described as an alternative to the MLP, but with "learnable activation functions". It relies on the Kolmogorov Representation Theorem to rewrite any multivariate function $f : [0,1]^n \mapsto \mathbb{R}$

$$f(x) = \sum_{i=1}^{2n+1} \Phi_i\left(\sum_{j=1}^{d} \phi_{ij}(\mathbf{x}_j)\right) \tag{2}$$

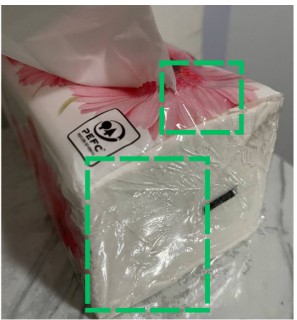

Figure 5: **Real-World Example of Specular Detail Sensitivity to Viewing Angle.** Small view angle change might result in large change in RGB

into a sum of univariate functions $\Phi_\square : [0,1] \mapsto \mathbb{R}$ and $\phi_\square : \mathbb{R} \mapsto \mathbb{R}$. For the vanilla KAN, the univariate functions are parameterized by B-Splines that have learnable coefficients, although KANs with alternative basis functions have also been recently implemented Aghaei (2024); Ta (2024); Seydi (2024); Yamamoto et al. (2020). For a network with $L$ layers the input-output mapping is as follows:

$$KAN(\mathbf{X}) = (\Phi_{L-1} \circ \Phi_{L-2} \cdots \Phi_1 \circ \Phi_0)(\mathbf{X}) \tag{3}$$

Due to the use of B-Splines, KANs exhibit a locality property where incoming data only affects the approximated function near the new input (Figure 4). This property makes it particularly suitable for modelling specular highlights which are highly sensitive to slight changes in view direction (Figure 5). In contrast, MLPs which are used in earlier works (Yang et al. (2024); Lu et al. (2023)) might be sensitive to these abrupt color shifts, leading to the phenomenon known as "catastrophic forgetting" where previously learnt information is lost while the network tries to fit the new incoming data.

### 3.1.3 ANISOTROPIC SPHERICAL GAUSSIAN (ASG).

ASGs are an alternative to the spherical gaussians utilized in vanilla 3DGS, which uses spherical harmonics to model the view dependent color of the gaussian. It is defined as the function:

$$ASG(\mathbf{v}; [\mathbf{x},\mathbf{y},\mathbf{z}], [\lambda,\mu], \eta) := \eta \cdot S(\mathbf{v}; \mathbf{z}) \cdot e^{-\lambda(\mathbf{v}\cdot\mathbf{x})^2 - \mu(\mathbf{v}\cdot\mathbf{y})^2} \tag{4}$$

where $\mathbf{x}$, $\mathbf{y}$ and $\mathbf{z}$ are the *lobe, tangent and bi-tangent axes* respectively; $\lambda, \mu \in \mathbb{R}$ are the bandwidths for $x$ and $y$ axes satisfying $\lambda, \mu > 0$, $\eta$ is the *lobe amplitude*, $S(\mathbf{v}; \mathbf{z}) = max(\mathbf{v} \cdot \mathbf{z}, 0)$ the *smooth term* and $e^{-\lambda(\mathbf{v}\cdot\mathbf{x})^2 - \mu(\mathbf{v}\cdot\mathbf{y})^2}$ the *exponential term*.

ASGs are rotational invariant (Xu et al. (2013)) and able to represent signals from all frequencies, making them effective at modelling high frequency details in the scene. Thus, by combining spherical harmonics and ASG (Yang et al. (2024)), a more expressive view-dependent color function is created by having spherical harmonics model low to mid frequency details while leaving high frequency details to ASG.

### 3.1.4 SPECTRAL LOSS.

Combining spectral loss with spatial loss is utilised in 3D reconstruction to improve quality (Yang et al. (2023)), and has yielded promising results when applied to 3DGS (Zhang et al. (2024)). For Phase-Aware KANGaussian, it begins with converting both the ground truth image $I \in \mathbb{R}^{H \times W \times C}$ and rendered image $\hat{I} \in \mathbb{R}^{H \times W \times C}$ from spatial domain to Fourier domain for each color channel $c$:

$$\mathcal{F}_c(u,v) = \frac{1}{\sqrt{HW}} \sum_{h=0}^{H-1} \sum_{w=0}^{W-1} I_c(h,w) \cdot e^{-i2\pi(u\frac{h}{H} + v\frac{w}{W})} \tag{5}$$

where $(u,v)$ and $(h,w)$ are coordinate values in frequency and spatial domain respectively. Then, we extract phase information:

$$\angle \mathcal{F}_c(u,v) = \arctan\left[\frac{Im(u,v)}{Re(u,v)}\right] \tag{6}$$

where $Im$ and $Re$ are the real and imaginary functions respectively. With the phase extracted for both $I$ and $\hat{I}$, we compute the channel-wise regularization loss:

$$\mathcal{L}_{c,\angle} = \frac{1}{\sqrt{HW}} \sum_{u=0}^{U} \sum_{v=0}^{V} \mid \angle\mathcal{F}_c(u,v) - \angle\hat{\mathcal{F}}_c(u,v) \mid \tag{7}$$

Lastly, we take the mean of the losses computed for the channels to get the final regularization term:

$$\mathcal{L}_{\angle} = \frac{1}{C} \sum_{c \in C} \mathcal{L}_{c,\angle} \tag{8}$$

## 3.2 PHASE-AWARE KANGAUSSIAN

### 3.2.1 ANCHOR-BASED GAUSSIAN SPLATTING.

Anchor-Based Gaussian Splatting (Lu et al. (2023)) starts with structure from motion (SfM), which generates a sparse point cloud $\mathbf{P} \in \mathbb{R}^{M \times 3}$. The point cloud would guide the initialization of a sparse voxel grid with voxel centres $\mathbf{V}$:

$$\mathbf{V} = \left\{ \left\lfloor \frac{\mathbf{P}}{\epsilon} \right\rfloor \right\} \cdot \epsilon \tag{9}$$

where $\mathbf{V}$ are the voxel centres and $\epsilon$ the voxel size. Duplicate entries are removed, denoted by $\{\cdot\}$

At each voxel centre, an anchor gaussian $v$ is generated, each with a set of $k$ learnable parameters including: offsets $\mathcal{O} \in \mathbb{R}^{k \times 3}$, scale $l_v \in \mathbb{R}^3$ and local context feature vector $\mathbf{f}_v \in \mathbb{R}^{32}$. The anchor gaussians would spawn $k$ offset gaussians, where their spatial positions (gaussian centre) are computed relative to the position of $v$:

$$\{\mathbf{x}_0, \ldots, \mathbf{x}_{k-1}\} = \mathbf{x}_v + \{\mathcal{O}_0, \ldots, \mathcal{O}_{k-1}\} \cdot l_v \tag{10}$$

where $\mathbf{x}_i$ is the center of the $i$-th neural Gaussian. Their opacity is derived from a compact MLP decoder $\mathcal{F}_\sigma$ (Lu et al. (2023)):

$$\{\sigma_0, \ldots, \sigma_{k-1}\} = \mathcal{F}_\sigma(\mathbf{f}_v, \delta_{cv}, \mathbf{d}_{cv}) \tag{11}$$

where $\delta_{cv}$ and $\mathbf{d}_{cv}$ are the distance and unit direction from the camera to the anchor gaussian respectively. Similarly, the rotation and scale of the offset gaussians are also derived this way, with a separate MLP decoder for each attribute.

Following Spec-Gaussian (Yang et al. (2024)), the feature vector $\mathbf{f}_v$ will be used to compress the $k$ ASGs. To recover the ASGs, in Phase-Aware KANGaussian, we instead utilize a KAN rather than a MLP as a decoder:

$$KAN(\mathbf{f}_v, \mathbf{d}_{co}) \rightarrow \{\lambda, \mu, \eta\}_k \tag{12}$$

where $\mathbf{d}_{co}$ is the unit direction vector from the camera to the offset gaussian.

As the initial point cloud is sparse, densification is necessary to ensure there are sufficient gaussians to represent the scene effectively. In Anchor-Based Gaussian Splatting, the space is segmented into a multi-resolution voxel grid to allow new anchor gaussians to be spawned at different granularity (Lu et al. (2023)):

$$\epsilon^{(m)} = \frac{\epsilon \cdot \beta}{4^m}, \tau_g^{(m)} = \tau_g \cdot 2^m \tag{13}$$

where $m$ is the level of new anchor gaussians, $\epsilon^{(m)}$ is the voxel size at the $m$-th level for newly spawned anchor gaussians, and $\beta$ is a growth factor. With training, the average gradients $\nabla_v$ and average opacity $\bar{\sigma}$ values of the offset gaussians will be accumulated. Then, only anchor gaussians with $\nabla_v > \tau_g$ and $\nabla_v > \text{Quantile}(\nabla_v, 2^{-(m+1)})$ will be densified at the corresponding voxel center at the $m$-th level, while those with $\bar{\sigma} < \tau_o$ will be removed, where $\tau_g$ and $\tau_o$ are predefined values.

### 3.2.2 ANISOTROPIC GAUSSIAN COLOR.

Phase-Aware KANGaussian utilizes a similar diffuse-specular color decomposition proposed by Spec-Gaussian (Yang et al. (2024)), where the anisotropic color $c$ of the gaussian is decomposed into two distinct components:

$$c = c_d + c_s \tag{14}$$

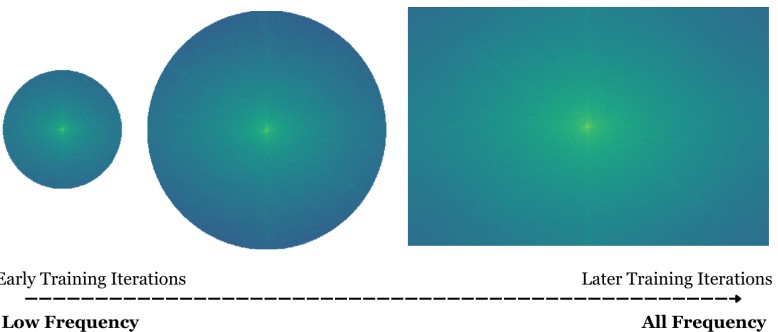

Early Training Iterations                                Later Training Iterations

**Low Frequency**                                        **All Frequency**

Figure 6: **Frequency Filtering with Training Iterations.** The expanding kernel only keeps low frequency information in early training iterations, eventually working up to include all frequencies after a certain iteration threshold

where $c_d$ and $c_s$ are the diffuse and specular components respectively. $c_d$ are represented by anisotropic spherical gaussians. Inspired by Spec-Gaussian, we form a basis with $N$ anisotropic spherical gaussians by initializing them with orthonormal axes $[\mathbf{x}, \mathbf{y}, \mathbf{z}]$ that are distributed evenly across a hemisphere. Then, when given a view direction, we can query the $N$ ASGs and concatenate the outputs to form a low dimensional representation of the high frequency color information:

$$\kappa = \bigoplus_{i}^{N} ASG(\mathbf{v}_r; [\mathbf{x}, \mathbf{y}, \mathbf{z}], [\lambda_i, \mu_i], c_i), \mathbf{v}_r = 2(\mathbf{v}_0 \cdot \mathbf{n}) \cdot \mathbf{n} - \mathbf{v}_0 \tag{15}$$

where $\mathbf{v}_0 = -\mathbf{d}$ is the unit direction from the spherical gaussian to the camera in world frame, and $\mathbf{n}$ being unit vector corresponding to the shortest axis of the spherical gaussian which is used to approximate the surface normal of the gaussian. To recover the specular color $c_s$, $\kappa$ is fed into the KAN decoder:

$$c_s = KAN(\kappa, \gamma(\mathbf{d}), \langle n, -\mathbf{d}\rangle) \tag{16}$$

where $\gamma$ is the positional encoding function.

### 3.2.3 EXPANDING KERNEL.

Phase-Aware KANGaussian utilizes a kernel $kern$ with an expanding frequency band, such that the frequency range $D_t$ at the $t \in [0, T']$ iteration allowed to pass is defined as:

$$D_t < D' + \frac{t}{T'}(D - D') \tag{17}$$

where $D'$ is a predefined frequency upper-threshold and $D$ is the maximum range of the frequency spectrum in the image. The expanding kernel ensures that the training begins with prioritizing low frequency details, building a strong foundation that high frequency details can be built upon. After obtaining $kern$, it is convolved with the image in Fourier domain to extract the necessary frequency range:

$$F' = F \circledast kern \tag{18}$$

where $\circledast$ is the convolution operator. The same kernel will be applied to both render and ground truth, before the spectral loss term $L_\angle$ is computed as illustrated in Figure 3.

### 3.2.4 CHOICE OF LOSS FUNCTION.

The loss function for Phase-Aware KANGaussian largely follows the loss functions of other Anchor-Based gaussian models (Lu et al. (2023); Yang et al. (2024)), with the addition of an additional phase regularization term $L_\angle$ introduced earlier:

$$\mathcal{L} = (1 - \lambda_{D-SSIM})\mathcal{L}_1 + \lambda_{D-SSIM}\mathcal{L}_{D-SSIM} + \lambda_{prod}\mathcal{L}_{prod} + \lambda_\angle\mathcal{L}_\angle \tag{19}$$

| Dataset | Mip-NeRF360 | | | Tanks&Temples | | | Deep Blending | | |
|---|---|---|---|---|---|---|---|---|---|
| Method — Metrics | PSNR ↑ | SSIM ↑ | LPIPS ↓ | PSNR ↑ | SSIM ↑ | LPIPS ↓ | PSNR ↑ | SSIM ↑ | LPIPS ↓ |
| Mip-NeRF360 | 27.69 | 0.792 | 0.237 | 0.06 | 0.759 | 0.257 | 29.40 | 0.901 | 0.245 |
| iNGP | 25.59 | 0.699 | 0.331 | 21.72 | 0.723 | 0.330 | 23.62 | 0.797 | 0.423 |
| Plenoxels | 23.08 | 0.626 | 0.463 | 21.08 | 0.719 | 0.379 | 23.06 | 0.795 | 0.510 |
| 3D-GS | 27.47 | 0.812 | 0.222 | 23.72 | 0.845 | 0.178 | 29.65 | 0.899 | 0.247 |
| Scaffold-GS | 27.98 | 0.807 | 0.236 | 23.96 | 0.853 | 0.177 | 30.21 | 0.906 | 0.254 |
| Spec-Gaussian | 28.00 | 0.819 | 0.205 | 24.54 | 0.857 | 0.175 | 30.34 | 0.909 | 0.253 |
| Ours | 27.96 | 0.822 | 0.186 | 24.47 | 0.862 | 0.161 | 30.32 | 0.911 | 0.231 |

Table 1: **Metric Scores of Our Method Compared to Previous Work on Real-World Datasets.** Cells are coloured according to best and second best .

and

$$\mathcal{L}_{prod} = \frac{1}{N} \sum_{i}^{N_n} Prod(\bar{s}_i) \tag{20}$$

where $N_n$ is the total number of offset gaussians, $Prod(\bar{s}_i)$ is the product of the scales $s_i$ of the $i$-th offset gaussian, and $\lambda_\square$ ($\square$ denotes subscripts in Equation 19) are scalar values to adjust the weights of the components of the loss function. The role of $L_{prod}$ is to keep the size of the offset gaussians small, such that overlap between the offset gaussians is minimized.

## 4 EXPERIMENTS

### 4.1 IMPLMENTATION DETAILS.

Phase-Aware KANGaussian is implemented on python on the Pytorch framework. The KAN decoder consists of 2 hidden layers, both with width 8.

The baseline for comparison is Spec-Gaussian (Yang et al. (2024)), which uses a Multilayer Perceptron to decode specualar features rather than a Kolmogorov-Arnold Network. Thus, we retained the original training parameters of SpecGaussian for the Anchor-Based Gaussian Splatting, namely: voxel size $\epsilon = 0.001$, densification threshold $\tau_g = 0.0002$, pruning threshold $\tau_o = 0.005$ and growth factor $\beta = 4$ for bounded scenes and $\beta = 16$ for Mip-NeRF360 scenes. Additionally, we also include the phase regularization term for Phase-Aware KANGaussian.

All experiments are trained and evaluated on $1 \times$ Nvidia A6000.

### 4.1.1 RESULTS & ANALYSIS.

From the experiments, Phase-Aware KANGaussian outperforms vanilla 3DGS and the baseline model in unbounded real-world scenes, especially in patches with high frequency details as seen in Figure 7. Our model encourages greater gaussian density at regions where high frequency information is congregated, allowing Phase-Aware KANGaussian to model the environment more accurately and generate better render quality.

From the perspective of metric scores referencing Table 1, Phase-Aware KANGaussian loses out in average PSNR across the three datasets. However, Phase-Aware KANGaussian significantly outperforms in other metrics, especially for LPIPS. This implies that the renders of Phase-Aware KANGaussian is able to model the high dimensional features of the scene better than the other models, but contains more noise than the other models, which is likely the byproduct of accumulated error in densified gaussian regions for modelling high frequency patches. It is also noteworthy that the KAN in Phase-Aware KANGaussian has a relatively lower model complexity ($8 \times 8$ hidden layers), as compared to the MLP utilized for SpecGaussian ($128 \times 128 \times 128$ hidden layers), which is also another factor to consider in the discrepancies in metric scores. The choice and impact of model complexity is further explored in the ablation study.

However, the KAN decoder poses a bottleneck for both training and inference speed, primarily due to the computational inefficiency of B-Splines, which have not yet been optimized for parallelization. This limitation reduces training and rendering speed of Gaussian Splatting, with Phase-Aware

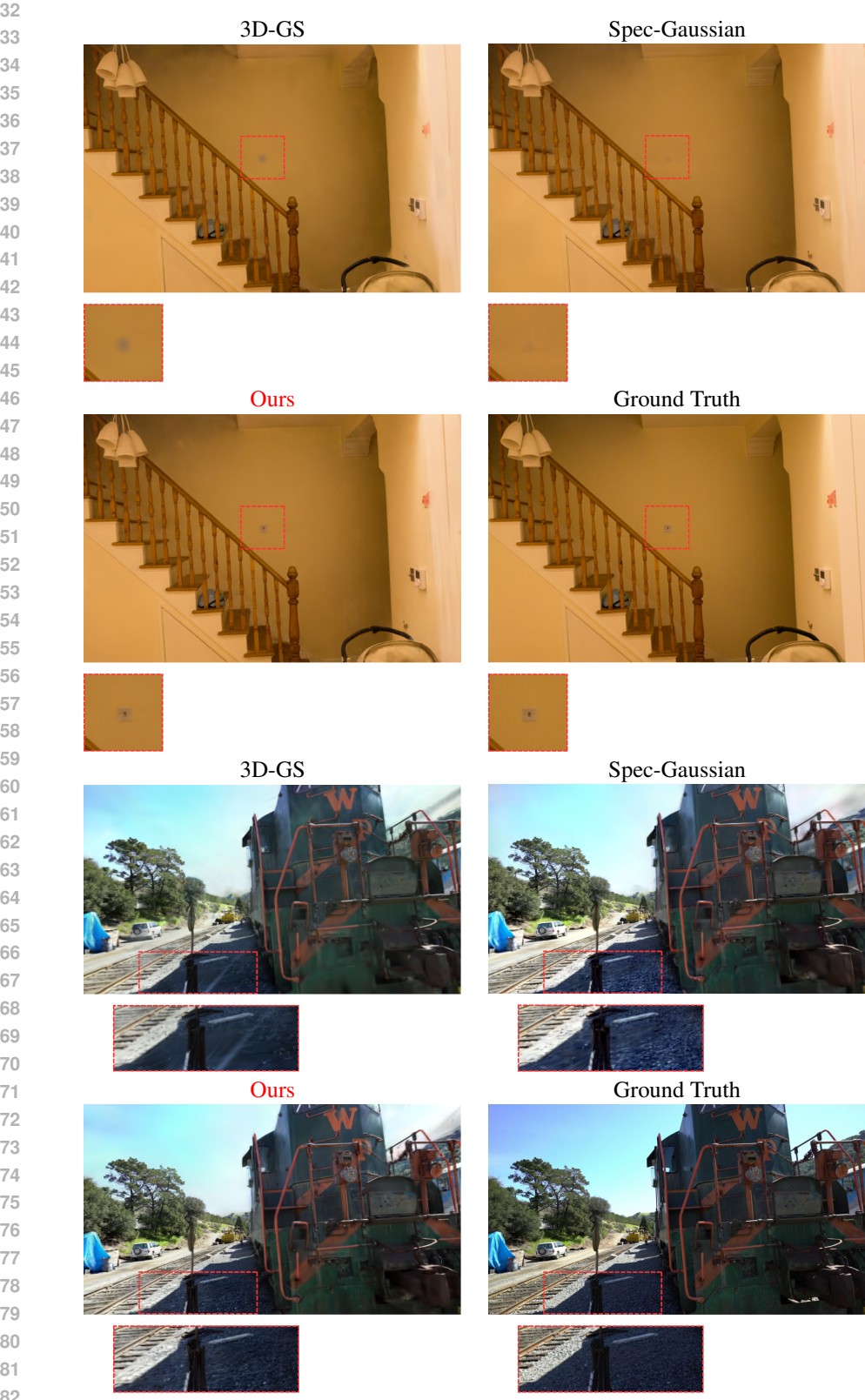

Figure 7: **Visualization of Improvements to High Frequency Patches.** Phase-Aware KANGaussian outperforming other models in modelling complex high frequency patches. **Zoom in for better visualization**

| Dataset | Mip-NeRF360 | | | Tanks&Temples | | | Deep Blending | | |
|---|---|---|---|---|---|---|---|---|---|
| Method — Metrics | PSNR ↑ | SSIM ↑ | LPIPS ↓ | PSNR ↑ | SSIM ↑ | LPIPS ↓ | PSNR ↑ | SSIM ↑ | LPIPS ↓ |
| No KAN | 27.80 | 0.824 | 0.185 | 24.33 | 0.862 | 0.159 | 30.28 | 0.909 | 0.234 |
| No Regularization (8 × 8 KAN) | 27.95 | 0.820 | 0.193 | 24.45 | 0.863 | 0.159 | 30.30 | 0.910 | 0.239 |
| Amplitude + Phase Regularization (8 × 8 KAN) | 27.81 | 0.825 | 0.185 | 24.39 | 0.862 | 0.159 | 30.22 | 0.910 | 0.232 |
| Phase Regularization (8 × 8 KAN) (Ours) | 27.96 | 0.822 | 0.186 | 24.47 | 0.862 | 0.161 | 30.32 | 0.911 | 0.231 |

Table 2: **Metric Scores of Our Method Compared with Ablation Study Models.** Cells are coloured according to best and second best .

| Dataset | Mip-NeRF360 | | | Tanks&Temples | | | Deep Blending | | |
|---|---|---|---|---|---|---|---|---|---|
| Method — Metrics | PSNR ↑ | SSIM ↑ | LPIPS ↓ | PSNR ↑ | SSIM ↑ | LPIPS ↓ | PSNR ↑ | SSIM ↑ | LPIPS ↓ |
| No Regularization (128 × 128 × 128 MLP) (Yang et al. (2024)) | 28.00 | 0.819 | 0.205 | 24.54 | 0.857 | 0.175 | 30.34 | 0.909 | 0.253 |
| No Regularization (8 × 8 KAN) | 27.95 | 0.820 | 0.193 | 24.45 | 0.863 | 0.159 | 30.30 | 0.910 | 0.239 |
| No Regularization (16 × 8 KAN) | 28.01 | 0.822 | 0.191 | 24.57 | 0.862 | 0.161 | 30.29 | 0.910 | 0.240 |

Table 3: **Metric Scores for Comparing Different KAN/MLP Model Complexities.** Cells are coloured according to best and second best .

KANGaussian being approximately four times slower in training and six times slower in rendering than SpecGaussian (Yang et al. (2024)). Future work on alternative basis functions for KANs and parallel computing optimizations could significantly enhance the efficiency of training and inference, making real-time rendering with Phase-Aware KANGaussian a more attainable goal.

### 4.2 ABLATION STUDY

#### 4.2.1 KOLMOGOROV-ARNOLD NETWORK DECODER

An ablation study removing the Kolmogorov-Arnold Network (KAN) component, recorded as "No KAN" in Table 2, reveals the complete removal of the specular component, leaving only diffuse colors in the render. This results in a significant drop in PSNR across datasets, with minor improvements in SSIM and LPIPS in some cases, though the PSNR drop is more pronounced.

Reconstruction quality is also influenced by KAN decoder complexity. We compare metric scores for an $8 \times 8$ and $16 \times 8$ hidden layer KAN, labeled as "No regularization ($8 \times 8$ KAN)" and "No regularization ($16 \times 8$ KAN)" in Tables 2 and 3. Despite being simpler than the $128 \times 128 \times 128$ MLP decoder in SpecGaussian (Yang et al. (2024)), the KAN library's high CUDA memory demands limit scalability. On the A6000 GPU, the $16 \times 8$ KAN fully depletes memory, even without regularization, while the $8 \times 8$ KAN is the feasible upper limit with regularization. Metric comparisons show better scores with the more complex KAN, indicating further potential constrained by implementation.

#### 4.2.2 PHASE REGULARIZATION

We assessed phase regularization through an ablation study using two additional models: (1) with amplitude and phase regularization and (2) without regularization, labeled as "Amplitude + Phase Regularization ($8 \times 8$ KAN)" and "No Regularization ($8 \times 8$ KAN)" in Table 2. Phase regularization alone significantly improves PSNR while only slightly affecting SSIM and LPIPS, demonstrating its effectiveness.

## 5 CONCLUSION

In conclusion, we introduce our Phase-Aware KANGaussian model, a 3DGS variation that uses an Anchor-Based method (Lu et al. (2023)) with anisotropic spherical gaussians (Yang et al. (2024)), combined with a Kolmogorov-Arnold Network decoder and a novel expanding kernel spectral loss computed from phase differences in Fourier domain. With these changes, we were able to extend the quality enhancements from earlier works beyond bounded synthetic scenes to generate state-of-the-art results in unbounded real-world scenes where geometry and details are less predictable. Our model effectively densify regions where high frequency details are abundant, which greatly increases the render quality of these high frequency patches. However, the accuracy of the newly generated gaussians in these regions still have room for improvement, which might be a possible area for further research. Nonetheless, Phase-Aware KANGaussian lays a solid foundation for further work to build upon, particularly for improving render quality of high frequency regions.

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

# 6 SUPPLEMENTARY VISUALIZATION

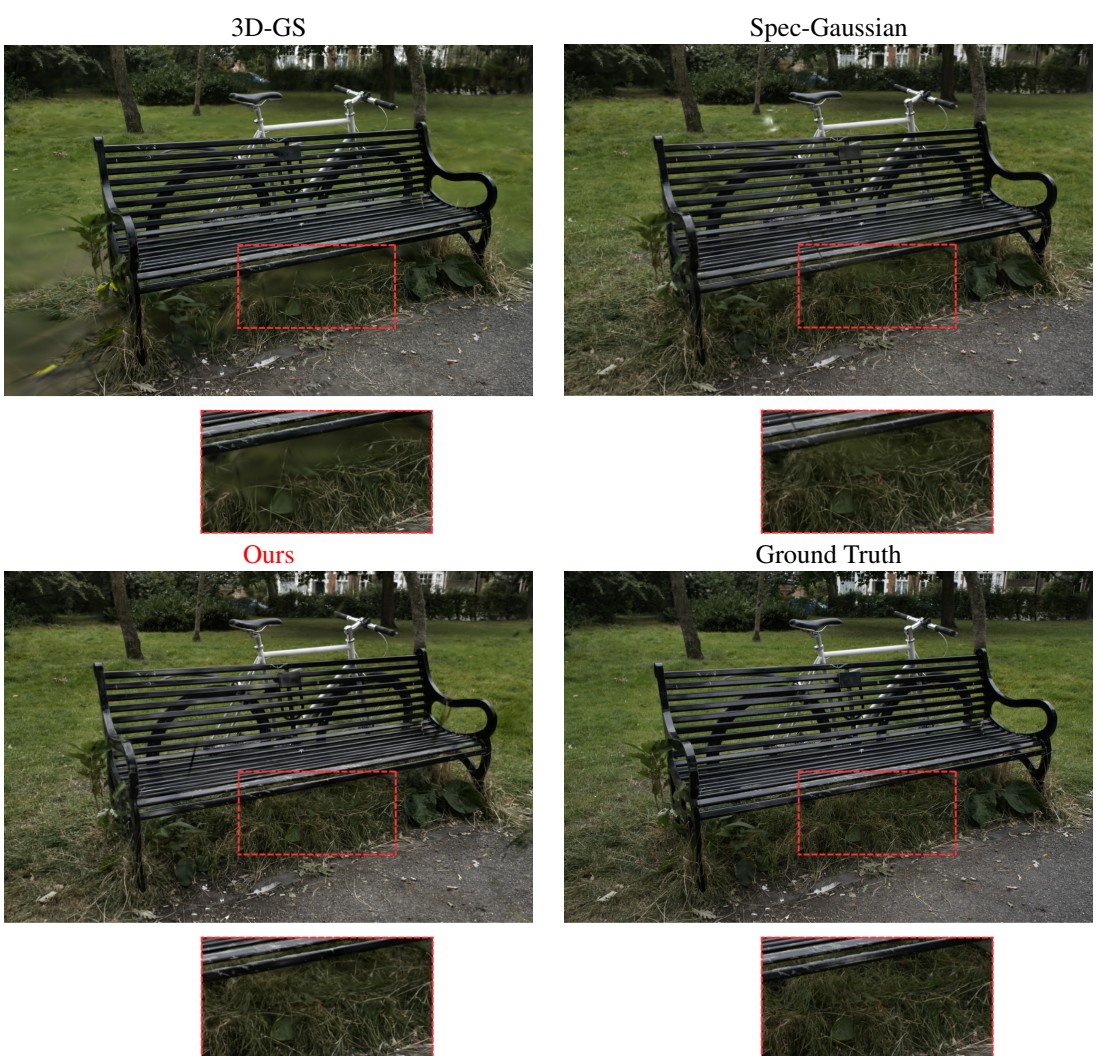

Figure 8: **Supplementary Visualization of Improvements to High Frequency Patches.** Phase-Aware KANGaussian outperforming other models in modelling complex high frequency patches. **Zoom in for better visualization**

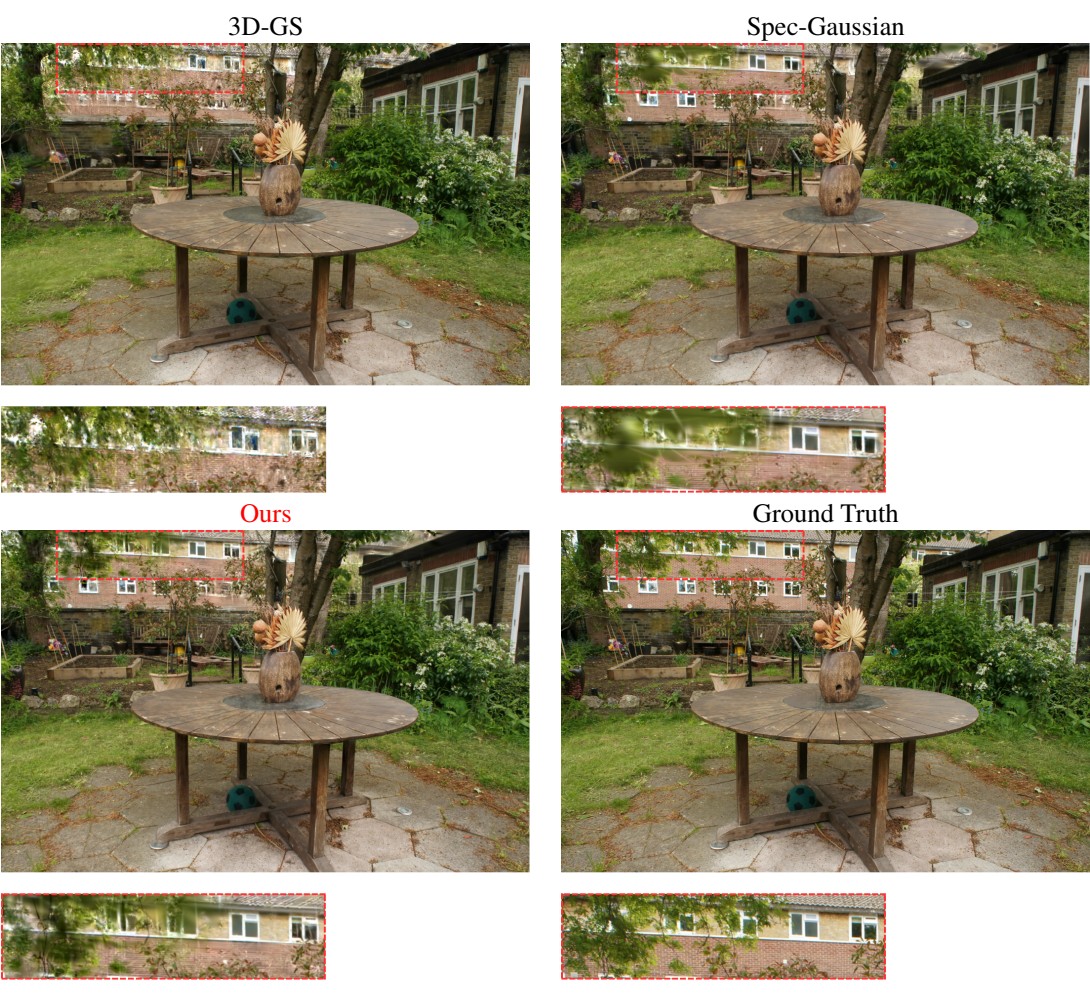

Figure 9: **More Supplementary Visualization of Improvements to High Frequency Patches.** Phase-Aware KANGaussian outperforming other models in modelling complex high frequency patches. **Zoom in for better visualization**

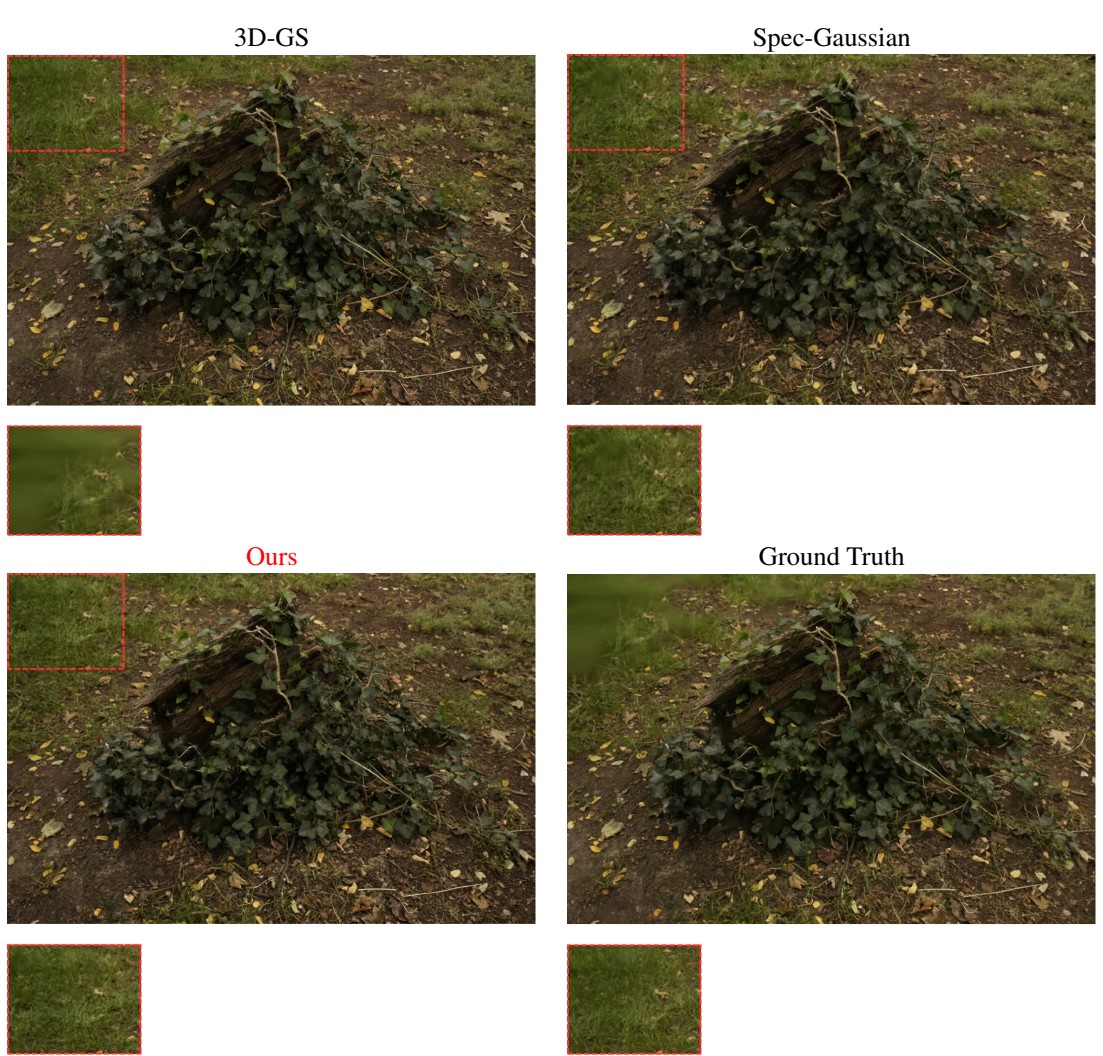

Figure 10: **Even More Supplementary Visualization of Improvements to High Frequency Patches.** Phase-Aware KANGaussian outperforming other models in modelling complex high frequency patches. **Zoom in for better visualization**

