# OpenReview forum: "Phase-Aware KANGaussian : Phase-Regularized 3D Gaussian Splatting with Kolmogorov-Arnold Network"
_ICLR.cc/2025/Conference — ICLR 2025 Conference Withdrawn Submission_

### Official Review · Reviewer_qU3P · 2024-10-16

**Soundness:** 3
**Presentation:** 3
**Contribution:** 2
**Rating:** 3
**Confidence:** 5

**Summary:**

The paper presents "Phase-Aware KANGaussian," a 3D reconstruction model that enhances the detail and quality of unbounded real-world scenes, particularly in high-frequency details. Its contributions can be summarised as:
1. Integrated 3D Gaussian Splatting with a Kolmogorov-Arnold Network (KAN) in the rendering procedure for improving rendering quality.
2. A phase regularization technique aimed at optimizing models from low to high frequency to dramatically enhance high-frequency detail rendering, which involves filtering before computing a regularization term in the Fourier domain.

**Strengths:**

1. The use of Kolmogorov-Arnold Networks in the 3DGS rendering pipeline is innovative, and the authors are the first few who are doing this.
2. The phase regularization approach for controlling frequency details during training could lead to more precise control over detail rendering in complex scenes.
3. The derivation of the formulas and the figures in the front part are used appropriately and clearly.
4. The problem is well stated.

**Weaknesses:**

1. The motivation for integrating KAN into 3DGS is not clear. How is the locality property of KANs expected to benefit the modeling of specular highlights and other high-frequency details?
2. The author put too much content in PRELIMINARIES and it feels like this article was cobbled together. Can you reorganize the preliminaries section to focus more tightly on the concepts most crucial to understanding the novel contributions?
3. There is a potential risk of overfitting to high-frequency details at the expense of overall scene fidelity, as indicated by the slightly lower PSNR scores compared to Spec-Gaussians.
4. The baseline (Spec-Gaussian (Yang, 2024)) of this article is not a peer-reviewed article, if there are peer-reviewed alternatives that could serve as additional comparisons?
5. The ablation study is confusing. e.g. Does "Phase Regularization (Ours)" contain Kan or does it only contain Phase Regularization? If does not, then which one is “No Kan” supposed to be compared to? Please provide a clear description of each ablation condition, including which components are present or absent in each case.

**Questions:**

1. The figure on Page 9 is too small. I have to zoom in "300%" to see it.
2. Please check whether the citation format is suitable for ICLR 2025. Sometimes the names of people are mixed with the sentences of the article, making it confusing. For example, "we employ Kolmogorov Arnold Networks (KANs) in the rendering pipeline in contrast to earlier works Lu et al. (2023)" -> "we employ Kolmogorov Arnold Networks (KANs) in the rendering pipeline in contrast to earlier works (Lu & Yu, 2023)"
3. Could you have some visual results for the ablation study? For example, some of your model's visual results remove some of your components.
4. Could you provide insights into the computational demands of your model, particularly regarding the use of KAN? (how much slower?)
5. How does the model perform under varied lighting conditions and metal areas, especially given its focus on high-frequency details which can be highly sensitive to such changes?
6. Could you elaborate on the potential causes for the observed decrease in PSNR?
7. There is an error in 3.2.4. "and λ□ are scalar values to adjust..."
8. What are the differences and advantages of your model over Mip-Splatting[1], which also focuses on high frequency?

[1] Yu, Z., Chen, A., Huang, B., Sattler, T. and Geiger, A., 2024. Mip-splatting: Alias-free 3d gaussian splatting. In Proceedings of the IEEE/CVF Conference on Computer Vision and Pattern Recognition (pp. 19447-19456).

---

> ### Author Response · Authors · 2024-12-02
> **Reply to Reviewer qU3P**
>
> We thank the reviewer for the suggestions.
>
> > The figure on Page 9 is too small. I have to zoom in "300%" to see it.
>
> We have revised the figure and moved some visualization samples into the supplementary part of the document.
>
> > Please check whether the citation format is suitable for ICLR 2025. Sometimes the names of people are mixed with the sentences of the article, making it confusing. For example, "we employ Kolmogorov Arnold Networks (KANs) in the rendering pipeline in contrast to earlier works Lu et al. (2023)" -> "we employ Kolmogorov Arnold Networks (KANs) in the rendering pipeline in contrast to earlier works (Lu & Yu, 2023)"
>
> We thank the reviewer for pointing the error in citation style out. We have revised the citation style accordingly.
>
> > Could you provide insights into the computational demands of your model, particularly regarding the use of KAN? (how much slower?)
>
> We have revised the paper to more clearly state the average slowdown of our model. Specifically, we see a 4x slowdown in training and 6x slowdown in rendering when compared against SpecGaussian. However, we hope for the reviewer's understanding that the KAN library is under optimized and has yet to be parallelized, leading to the discrepancy in computation time.
>
> > Could you elaborate on the potential causes for the observed decrease in PSNR?
>
> The KAN we have utilized in our model is only a 8x8 hidden layer KAN, which is less complex than the 128x128x128 MLP used in SpecGaussian. As such, the difference in PSNR is likely due to the model complexity, and we have further conducted an ablation study to verify this claim. We removed the frequency regularization component to free up CUDA memory to train a slightly more complex 16x8 hidden layer KAN model, and the results are tabulated below. We can observe a general improvement in metric scores for the 16x8 KAN model, highlighting the potential of KANs limited by implementation.
>
> NOTE: The models tabulated below do **not** have frequency regularization for fair comparison.
> |Model|Mip360 PSNR|Mip360 SSIM|Mip360 LPIPS|T&T PSNR|T&T SSIM|T&T LPIPS|DB PSNR|DB SSIM|DB LPIPS
> |-|-|-|-|-|-|-|-|-|-|
> |128x128x128 MLP (SpecGaussian)|28.00|0.819|0.205|24.54|0.857|0.175|**30.34**|0.909|0.253
> |8x8 KAN|27.95|0.820|0.193|24.45|**0.863**|**0.159**|30.30|**0.910**|**0.239**
> |16x8 KAN|**28.01**|**0.822**|**0.191**|**24.57**|0.862|0.161|30.29|**0.910**|0.240
>
> > There is an error in 3.2.4. "and λ□ are scalar values to adjust..."
>
> We apologize for the confusion, and have revised the paper to improve readability.
>
> > The ablation study is confusing. e.g. Does "Phase Regularization (Ours)" contain Kan or does it only contain Phase Regularization? If does not, then which one is “No Kan” supposed to be compared to? Please provide a clear description of each ablation condition, including which components are present or absent in each case.
>
> We apologize for the confusion again. We have revised the paper accordingly to be clearer in the ablation condition.
>
> > The baseline (Spec-Gaussian (Yang, 2024)) of this article is not a peer-reviewed article, if there are peer-reviewed alternatives that could serve as additional comparisons?
>
> The Spec-Gaussian paper has since been peer-reviewed and accepted into NeurIPS after the submission of our paper.
>
> > How does the model perform under varied lighting conditions and metal areas, especially given its focus on high-frequency details which can be highly sensitive to such changes?
>
> We thank the reviewer for the suggestion. We have attempted to run the model on the synthetic dataset of SpecGaussian[R2] that was made public after the submission of our paper, in order to simulate the varied lighting conditions and metallic reflection. Unfortunately, COLMAP fails on the dataset and we apologize for not being able to produce metric scores within the timespan of the rebuttal phase, due to the lengthy training process required for the KAN network.
>
> [R2] Yang, Ziyi, et al. "Spec-gaussian: Anisotropic view-dependent appearance for 3d gaussian splatting." arXiv preprint arXiv:2402.15870 (2024).

---

### Official Review · Reviewer_dgXq · 2024-10-19

**Soundness:** 3
**Presentation:** 3
**Contribution:** 3
**Rating:** 5
**Confidence:** 5

**Summary:**

This paper aims to apply KAN within the 3DGS framework to achieve higher-quality rendering. The authors primarily build upon Spec-Gaussian, Scaffold-GS, and Fre-GS, replacing the MLP with KAN, which results in improved rendering quality. Additionally, phase regularization is introduced to further enhance visual results. The experimental results show that KANGaussian achieves impressive results in real-world scenarios.

**Strengths:**

- This paper is well-written and easy to understand.
- The use of KAN in Gaussian splatting is novel.
- The impressive visual results and ablation studies are appreciated.

**Weaknesses:**

1. This paper lacks sufficiently novel methods. For example, Sections 3.2.1 and 3.2.2 are largely based on Spec-Gaussian (with the exception of differences in KAN and MLP). Section 3.2.3, on the other hand, is based on Fre-GS. The paper seems to be a KAN version that combines Spec-Gaussian with Fre-GS. I recommend that the authors move parts of these sections to the Preliminary section.
2. The authors dedicated a significant portion of the paper to explaining how KANGaussian theoretically offers a higher capacity for modeling high-frequency information. However, in the Experiments section, there are no examples that demonstrate improvements in modeling specular components; instead, the focus is on floater removal, as seen in Figure 7. It could be much better if the authors used scenes with specular highlights, such as the example shown in Figure 5, to substantiate their claims and prove the effectiveness of their method in improving specular modeling.
3. Missing comparison of training time and rendering efficiency (FPS). There is still a computational speed difference between KAN and MLP. Although KANGaussian may not have an advantage in rendering speed, it could be much better to provide these details to give readers a clearer understanding of the strengths and weaknesses of the KAN-based approach.
4. Missing comparison of the number of Gaussians. The quantity of Gaussians has a significant impact on rendering metrics, and the authors need to provide a comparison of the actual number of Gaussians used in each method to ensure a fair comparison.

**Questions:**

I'm very curious about the resolution the authors used for Mip-NeRF 360. Did they follow the Mip-360 setup with downsampling factors of 4 for outdoor scenes and 2 for indoor scenes, or did they adopt the 3DGS setting where the images are uniformly cropped to a width of 1600 pixels?

---

> ### Author Response · Authors · 2024-12-02
> **Reply to Reviewer dgXq**
>
> We thank the reviewer for their constructive feedback on our paper.
>
> > This paper lacks sufficiently novel methods. For example, Sections 3.2.1 and 3.2.2 are largely based on Spec-Gaussian (with the exception of differences in KAN and MLP). Section 3.2.3, on the other hand, is based on Fre-GS. The paper seems to be a KAN version that combines Spec-Gaussian with Fre-GS. I recommend that the authors move parts of these sections to the Preliminary section.
>
> We understand the reviewer's concerns about our paper's novelty. However, the choice of KAN is intentional to exploit the locality property of the network to better model specular reflections that are highly view-dependent. There is great potential in KANs for this task as we demonstrate that using a KAN with 16x8 hidden layers can generate comparable metric scores to a MLP with 128x128x128 hidden layers (used in SpecGaussian) in our ablation study. Unfortunately, as KANs are new, the main bottleneck is the under-optimized implementation as the computation of the B-Splines are yet to be parallelized. We hope that our work can spark more interest in KANs, hopefully leading to better optimized libraries that can generate new groundbreaking results.
>
> With regards to frequency regularisation, incorporating an expanding mask into the regularization process have been widely utilized in other applications of 3D computer vision, including in NeRFs (FreeNERF[R1]). However, as far as we are aware, the approach of only including the phase term in the regularization process is novel. The motivation behind excluding the amplitude term is that spatial information is mostly contained in the phase, so including (and optimizing for) the amplitude term might dilute the effectiveness of the regularization on the reconstruction. This is verified in the ablation study, where we compared the reconstruction quality when we included both terms against only including the phase term, and our method yielded better metric scores in general.
>
> Lastly, with regards to the organization of the paper, we hope for the understanding of the reviewer that the sections have proper citations in their body paragraphs and they are crucial to the rendering pipeline of the method, so moving the suggested sections to the preliminary section might hinder readability and flow of the paper.
>
> [R1] Yang, J., Pavone, M., & Wang, Y. (2023). Freenerf: Improving few-shot neural rendering with free frequency regularization. In Proceedings of the IEEE/CVF conference on computer vision and pattern recognition (pp. 8254-8263).
>
> > The authors dedicated a significant portion of the paper to explaining how KANGaussian theoretically offers a higher capacity for modeling high-frequency information. However, in the Experiments section, there are no examples that demonstrate improvements in modeling specular components; instead, the focus is on floater removal, as seen in Figure 7. It could be much better if the authors used scenes with specular highlights, such as the example shown in Figure 5, to substantiate their claims and prove the effectiveness of their method in improving specular modeling.
>
> We thank the reviewer for the suggestion. We are currently working to include additional metric scores for specular synthetic dataset and would ask for the reviewer's patience.
>
> > Missing comparison of training time and rendering efficiency (FPS). There is still a computational speed difference between KAN and MLP. Although KANGaussian may not have an advantage in rendering speed, it could be much better to provide these details to give readers a clearer understanding of the strengths and weaknesses of the KAN-based approach.
>
> We agree with the reviewer and have revised the paper to include the average slowdown in training time (4x slowdown vs SpecGaussian) and slowdown in rendering time (6x slowdown vs SpecGaussian). However, we also seek the reviewer's understanding in that KANs are new so the libraries are under optimized, thus the updating of parameters for KANs have yet to be parallelized.
>
> > Missing comparison of the number of Gaussians. The quantity of Gaussians has a significant impact on rendering metrics, and the authors need to provide a comparison of the actual number of Gaussians used in each method to ensure a fair comparison.
>
> The number of gaussians remain unaffected with the introduction of KAN. However, there is a 2x increase in number of gaussians when frequency regularization is applied, as it encourages more gaussians to be produced in order to model high frequency details.
>
> > I'm very curious about the resolution the authors used for Mip-NeRF 360. Did they follow the Mip-360 setup with downsampling factors of 4 for outdoor scenes and 2 for indoor scenes, or did they adopt the 3DGS setting where the images are uniformly cropped to a width of 1600 pixels?
>
> We utilized the Mip-360 setup as described in the question.

---

### Official Review · Reviewer_eqJD · 2024-11-02

**Soundness:** 3
**Presentation:** 3
**Contribution:** 3
**Rating:** 6
**Confidence:** 4

**Summary:**

This study investigates integrating KAN into the 3DGS framework to enhance rendering quality. By replacing MLP with KAN in Neural-Appearance GS techniques like Scaffold-GS, the authors achieve improved visual outcomes. Phase regularization is applied to further refine the visuals, leading to satisfactory results. However, the approach is somewhat limited, as it combines KAN and GS directly on color prediction, serving as a continuation of Scaffold-GS and Spec-GS.

**Strengths:**

1. The paper excels in presenting a novel integration of KAN within the 3DGS framework, leading to significant improvements in rendering quality. The authors effectively demonstrate how replacing MLP with KAN in established methods like Spec-Gaussian and Fre-GS enhances visual outcomes. This innovative approach not only improves the clarity and detail of rendered images but also introduces phase regularization to refine the results further.

2. The paper is well-organized and clearly written, making complex concepts accessible to readers. The authors provide comprehensive ablation studies and figure illustrations that thoroughly support their claims, showcasing the superiority of KANGaussian over traditional methods in real-world scenarios. The method's potential to handle high-frequency details and improve visual fidelity is well-articulated, backed by detailed experimental results that highlight its practical applicability and robustness.

**Weaknesses:**

1. There is no comparison of training time and rendering speed. One of Gaussian's greatest advantages is its fast rendering and minimal training time. Including quantitative measurements of training and inference time would clarify KAN's impact on GS.

2. As mentioned in the summary, I find the direct combination of KAN and GS in the well-explored area of neural GS appearance to be somewhat trivial. However, I believe that experimenting with novel technique combinations and sharing results benefits the community. I encourage such efforts, especially when the technique is straightforward. The results, though, are not significantly superior to other methods.

**Questions:**

I hope the author reports the training and inference speeds, as these are two of GS's greatest advantages and are of significant interest to readers.

---

> ### Author Response · Authors · 2024-12-02
> **Reply to Reviewer eqJD**
>
> We appreciate the reviewer's constructive feedback.
>
> > There is no comparison of training time and rendering speed. One of Gaussian's greatest advantages is its fast rendering and minimal training time. Including quantitative measurements of training and inference time would clarify KAN's impact on GS.
>
> We acknowledge the reviewer's concerns and have included in the revised paper the average slowdown in training (4x slower vs SpecGaussian) and inference time (6x slower vs SpecGaussian), as well as explanations behind the slowdown, primarily being that the KAN library is relatively new and computation of the basis functions for KAN has yet to be parallelised.

---

### Official Review · Reviewer_HoeG · 2024-11-03

**Soundness:** 2
**Presentation:** 2
**Contribution:** 2
**Rating:** 3
**Confidence:** 5

**Summary:**

This paper presents a 3DGS method Phase-Aware KANGaussian method, which aims to enhance 3D reconstruction quality, particularly for capturing high-frequency details in unbounded real-world scenes. The authors propose a novel phase regularization technique that progressively optimizes model training across frequencies from low to high. Additionally, they integrate the Kolmogorov-Arnold Network (KAN) into anisotropic color modeling.

**Strengths:**

The introduction of KAN for modeling anisotropic color is novel. It might be theoretically better than MLP, as KAN exhibits a locality property due to its B-Splines as claimed in Line 245-246 and Figure 4.

**Weaknesses:**

## Major Concerns:
1. The novelty of phase regularization is questionable. FreGS [R1] has already provided frequency regularization on both amplitudes and phase parts. Equation 7 in this paper is similar to Equation 6 in FreGS. Besides, this paper introduces frequency filtering by expanding the frequency band, which is similar to the frequency annealing proposed in FreGS. For example, Equation 17 in this paper is similar to Equation 13 in FreGS. Please clarify the difference from FreGS, particularly for the above two aspects. Also, a comprehensive experimental comparison to validate the superior advantage of the proposed method is necessary.

[R1] Zhang, Jiahui, et al. "Fregs: 3d gaussian splatting with progressive frequency regularization." Proceedings of the IEEE/CVF Conference on Computer Vision and Pattern Recognition. 2024.

2. The lack of evaluation on synthetic datasets. Through the authors' claim of their SOTA performance on real unbounded scenes, they should also validate the proposed method on synthetic shiny scenes as used in Spec-Gaussian [R2] and [R3].

[R2] Yang, Ziyi, et al. "Spec-gaussian: Anisotropic view-dependent appearance for 3d gaussian splatting." arXiv preprint arXiv:2402.15870 (2024).
[R3] Ye, Keyang, Qiming Hou, and Kun Zhou. "3d gaussian splatting with deferred reflection." ACM SIGGRAPH 2024 Conference Papers. 2024.

3. The experimental results presented in Table 1 for comparisons with Scaffold-GS [R4] raise some concerns. Specifically, the performance of Scaffold-GS on Mip-NeRF 360 is notably lower than the values reported in the original paper, whereas results on the other two datasets are the same. The authors do not clarify whether these results are based on retrained models or reporting values from the original work. Thus, the validity of the conclusions drawn from these comparisons is unclear. Please provide a more detailed description of the experimental setup and an explanation for the observed discrepancies in these results.

[R4] Lu, Tao, et al. “Scaffold-GS: Structured 3d gaussians for view-adaptive rendering.” Proceedings of the IEEE/CVF Conference on Computer Vision and Pattern Recognition. 2024.

4. The presented experimental results are not fully convincing. For instance, in the overall comparison of real datasets (Table 1), the proposed method ranks second in PSNR, underperforming compared to Spec-Gaussian [R2]. Additionally, in the ablation studies, the “No KAN” variant surprisingly outperforms the proposed method on Mip-NeRF 360 and Tanks&Temples in SSIM and LPIPS. Given SSIM and LPIPS’ importance in assessing texture detail, more thorough explanations and additional experiments are needed to validate the effectiveness of each module.

5. A direct comparison between the Kolmogorov-Arnold Network (KAN) and the MLP used in Spec-Gaussian [R2] is missing. Since the experimental results do not consistently surpass Spec-Gaussian on PSNR (Table 1), further evidence is needed to substantiate the choice of KAN over MLP.

6. The hyperparameters are not provided, e.g., the scalar terms of production of scale and phase regularization (Equation 19) used in experiments. Besides, an analysis or explanation of hyperparameter choice is better to be provided.

7. Certain aspects of the writing lack clarity and structural coherence, hindering readability and comprehension of the paper’s innovations. Here are some examples:
* Lines 253-254: Potential confusion between “spherical Gaussians” and “spherical harmonics.”
* Lines 259-260: Grammar issues in explaining the smooth and exponential terms.
* Line 402: Mislabeling “\lambda_{prod}” as “\lambda_{}”.


## Suggestions
1. A visualization of the ablation studies would offer clearer insights into the contributions of each component.

2. A more detailed comparison of KAN and MLP is recommended, in addition to the accuracy, evaluations of time efficiency and resource consumption are also required.


In summary, while this paper incorporates KAN into the 3DGS framework, the highlighted weaknesses, such as less novelty, lack of comparison on shiny datasets, unclear comparisons, suboptimal experimental results, insufficient justification of KAN, and limited validation, need to be addressed.

**Questions:**

see above

---

> ### Author Response · Authors · 2024-11-26
> **Response to Reviewer HoeG (Part 1)**
>
> We appreciate and thank the reviewer for their constructive feedback. We kindly request for the reviewer's patience as we are running experiments to answer the reviewer's queries, and revising the paper with the given feedback.
>
> >The novelty of phase regularization is questionable. FreGS [R1] has already provided frequency regularization on both amplitudes and phase parts. Equation 7 in this paper is similar to Equation 6 in FreGS. Besides, this paper introduces frequency filtering by expanding the frequency band, which is similar to the frequency annealing proposed in FreGS. For example, Equation 17 in this paper is similar to Equation 13 in FreGS. Please clarify the difference from FreGS, particularly for the above two aspects. Also, a comprehensive experimental comparison to validate the superior advantage of the proposed method is necessary.
>
> Frequency regularization with expanding masks have been widely utilized in other applications of 3D computer vision, including in NeRFs (FreeNERF[R1]). However, as far as we are aware, the approach of only including the phase term in the regularization process is novel. The motivation behind excluding the amplitude term is that spatial information is mostly contained in the phase, so including (and optimizing for) the amplitude term might dilute the effectiveness of the regularization on the reconstruction. This is verified in the ablation study, where we compared the reconstruction quality when we included both terms against only including the phase term, and our method yielded better metric scores in general.
>
> With regards to experimental comparison, we are unable to compare our proposed method against FreGS’ method directly as the separation of low and high frequency regularization losses is incompatible with the training pipeline of the Neural Gaussian Splatting approach. Specifically, the neural approach involves finding the subset of neural gaussians that need to be trained for a specific training image, but the low frequency components often do not contain sufficient spatial information to identify the subset (leading to an empty set of neural gaussians identified), which crashes the code during the backward pass. Therefore, our next best alternative was to use only one frequency loss term with both amplitude and phase (akin to setting the same weights for both frequency regularization terms for FreGS) and compare the reconstruction against omitting the amplitude term, which were exactly the models used in the ablation study.
>
> [R1] Yang, J., Pavone, M., & Wang, Y. (2023). Freenerf: Improving few-shot neural rendering with free frequency regularization. In Proceedings of the IEEE/CVF conference on computer vision and pattern recognition (pp. 8254-8263).
>
> > A direct comparison between the Kolmogorov-Arnold Network (KAN) and the MLP used in Spec-Gaussian [R2] is missing. Since the experimental results do not consistently surpass Spec-Gaussian on PSNR (Table 1), further evidence is needed to substantiate the choice of KAN over MLP.
>
>  As KANs are new, their libraries are under optimized compared to MLP libraries. As such, we were only able to fit a relatively shallow KAN model with 8x8 hidden layers (vs SpecGaussian’s 128x128x128 hidden layers) with our frequency regularization as any larger would run out of CUDA memory. However, by excluding the frequency regularization entirely to save CUDA memory, we were able to fit a more complex 16x8 KAN model, and the performance metrics are summarized in the table below. We observe that we are indeed able to extract better metric scores with a more complex KAN, indicating untapped potential limited by implementation. Thus, we hope that our work can inspire more researchers to incorporate KANs into their work, which would motivate better optimized KAN libraries to facilitate new breakthroughs.
>
> NOTE: The models tabulated below do **not** have frequency regularization for fair comparison.
> |Model|Mip360 PSNR|Mip360 SSIM|Mip360 LPIPS|T&T PSNR|T&T SSIM|T&T LPIPS|DB PSNR|DB SSIM|DB LPIPS
> |-|-|-|-|-|-|-|-|-|-|
> |128x128x128 MLP (SpecGaussian)|28.00|0.819|0.205|24.54|0.857|0.175|**30.34**|0.909|0.253
> |8x8 KAN|27.95|0.820|0.193|24.45|**0.863**|**0.159**|30.30|**0.910**|**0.239**
> |16x8 KAN|**28.01**|**0.822**|**0.191**|**24.57**|0.862|0.161|30.29|**0.910**|0.240

---

> ### Author Response · Authors · 2024-12-02
> **Response to Reviewer HoeG (Part 2)**
>
> > The lack of evaluation on synthetic datasets. Through the authors' claim of their SOTA performance on real unbounded scenes, they should also validate the proposed method on synthetic shiny scenes as used in Spec-Gaussian [R2] and [R3].
>
> We agree with the reviewer. At the time of writing the first version of our paper, the synthetic shiny dataset was not released by the authors of Spec-Gaussian. However, they have since made their dataset public, and we have been attempting to run our model on the shiny dataset in Spec-Gaussian[R2], but COLMAP fails on the dataset and hence we are unable to extract metric scores from the dataset.
>
> [R2] Yang, Ziyi, et al. "Spec-gaussian: Anisotropic view-dependent appearance for 3d gaussian splatting." arXiv preprint arXiv:2402.15870 (2024). [R3] Ye, Keyang, Qiming Hou, and Kun Zhou. "3d gaussian splatting with deferred reflection." ACM SIGGRAPH 2024 Conference Papers. 2024.
>
> > The presented experimental results are not fully convincing. For instance, in the overall comparison of real datasets (Table 1), the proposed method ranks second in PSNR, underperforming compared to Spec-Gaussian [R2]. Additionally, in the ablation studies, the “No KAN” variant surprisingly outperforms the proposed method on Mip-NeRF 360 and Tanks&Temples in SSIM and LPIPS. Given SSIM and LPIPS’ importance in assessing texture detail, more thorough explanations and additional experiments are needed to validate the effectiveness of each module.
>
> We understand the concerns of the reviewer, and agree with the analysis about the importance of SSIM and LPIPS in assessing textural detail. For the "No KAN' variant, we remove the specular highlights completely. Thus, the SSIM and LPIPS scores are unaffected by errors from modelling the specular highlights. As such, there might be minor discrepancies in SSIM and LPIPS scores, which is less significant in comparison to the difference in PSNR score.
>
> > A direct comparison between the Kolmogorov-Arnold Network (KAN) and the MLP used in Spec-Gaussian [R2] is missing. Since the experimental results do not consistently surpass Spec-Gaussian on PSNR (Table 1), further evidence is needed to substantiate the choice of KAN over MLP.
>
> We appreciate the reviewer's feedback. We have added an additional Table 3 in the ablation study to provide the direct comparison. Additionally, we hope for the reviewer's understanding in that the KAN network used is less complex than the MLP used in SpecGaussian, which could possibly affect the discrepancy in metric scores.
>
> > The hyperparameters are not provided, e.g., the scalar terms of production of scale and phase regularization (Equation 19) used in experiments. Besides, an analysis or explanation of hyperparameter choice is better to be provided.
>
> For Phase-Aware KANGaussian, through hyperparamter tuning, we found that choosing the scalar terms such that the regularization loss accounts for around 20% of the total loss (~10^-5) yielded the best results for the the three datasets we experimented on, as the regularization would perform more of a supportive function rather than overwhelm the loss function. However, the optimal hyperparameter is closely related to the specific dataset as the distribution of phase signals in each dataset might be different.
>
> > Certain aspects of the writing lack clarity and structural coherence, hindering readability and comprehension of the paper’s innovations.
>
> We thank the reviewer for their comments and have revised the paper to hopefully improve the readability and comprehension.

---

### Note · Authors · 2025-02-01

I have read and agree with the venue's withdrawal policy on behalf of myself and my co-authors.